# Tissue Bioconcentration Pattern and Biotransformation of Per-Fluorooctanoic Acid (PFOA) in *Cyprinus carpio* (European Carp)—An Extensive In Vivo Study

**DOI:** 10.3390/foods12071423

**Published:** 2023-03-27

**Authors:** Valentina Andreea Petre, Florentina Laura Chiriac, Irina Eugenia Lucaciu, Iuliana Paun, Florinela Pirvu, Vasile Ion Iancu, Laura Novac, Stefania Gheorghe

**Affiliations:** National Research and Development Institute for Industrial Ecology—ECOIND, Drumul Podu Dambovitei 57-73, Sector 6, 060652 Bucharest, Romania

**Keywords:** PFOA, bioconcentration, biotransformation, *Cyprinus carpio*

## Abstract

The perfluoroalkyl substances (PFAS) represent a persistent class of synthetic chemicals that spread in the environment as a result of industrialization. Due to their bioaccumulative and endocrine disruption implications, these chemicals can affect food quality and human health, respectively. In the present study, the bioconcentration and biotransformation of perfluorooctanoic acid (PFOA) in common carp (*Cyprinus carpio*) were evaluated in a biphasic system (exposure and depuration). Carp were continuously exposed, under laboratory conditions, to 10 (Experiment 1) and 100 (Experiment 2) µg/L PFOA for 14 weeks, followed by a wash out period of 3 weeks. Fish organs and tissues were collected at 8, 12, 14 weeks of exposure and at week 17, after the depuration period. The results obtained from the LC-MS/MS analysis showed the presence of PFOA in all studied organs. The highest values of PFOA were identified in the gallbladder (up to 2572 ng/g d.w.) in Experiment 1 and in the gallbladder (up to 18,640 ng/g d.w.) and kidneys (up to 13,581 ng/g d.w.) in Experiment 2. The average BCF varied between 13.4 and 158 L/Kg in Experiment 1 and between 5.97 and 80.3 L/Kg in Experiment 2. Four biotransformation products were identified and quantified in all organs, namely: PFBA, PFPeA, PFHxA, and PFHpA. PFBA was proven to be the dominant biotransformation product, with the highest values being determined after 8 weeks of exposure in the kidney, gallbladder, brain, liver, and gonads in both experiments. Because freshwater fish are an important food resource for the human diet, the present study showed the fishes’ capacity to accumulate perfluoroalkyl substances and their metabolites. The study revealed the necessity of monitoring and risk studies of new and modern synthetic chemicals in aquatic resources.

## 1. Introduction

PFAS (perfluoroalkyl substances) are a large class of synthetic chemical compounds, structurally characterized by a chain of aliphatic carbon atoms in which the hydrogen atoms are completely (per-) or partially (poly-) replaced by fluorine [1]. Due to their structural characteristics, PFAS possess high polarity, thermal, and chemical stability, which makes them suitable for a wide range of applications, such as surfactants in textile surface coatings, soil repellents, food packaging, cleaning products, cosmetics, medical devices, pesticides, and firefighting foams [2]. Consequently, they become persistent and difficult to manage in various environments, such as soil [3], water [4,5], air [6], sediment [7,8,9], surface water [7,9,10,11], rainwater [12], snow [12,13], wastewater [10,14,15], sewage sludge [10], food [16,17,18], biota [10,11], and even in the human body [1,19,20,21].

Among the existing classes of PFAS in recently published studies, the emphasis was placed on the detection and quantification of perfluoroalkyl carboxylic acids (PFCA, C_n_F_2n_-COOH, *n* ≥ 7) and on their derivatives [22]. From this class of substances, PFOA is the most studied representative, due to the level of use and implicitly the frequency of detection in the environment. PFOA has been proven to be bioaccumulative, toxic, persistent, and, more importantly, one of the most well-known endocrine disruptors, along with BPA (bisphenol A) [23]. As an endocrine disruptor, PFOA is able to interfere with estrogen and thyroid hormone receptors. Using human adrenocortical carcinoma cell lines, it was observed that PFOA increased the serum estradiol production and decreased the serum testosterone production, altered the expression of major steroidogenic genes, and regulated steroidogenic factors [24].

PFOA is released into the environment through industrial production and ends up in the food chain, similar to the rest of the persistent pollutants. The main ways by which PFOA can accumulate in human bodies are food (fish and seafood, meat and meat products, eggs, milk, and dairy products) [25,26,27] and through water sources [4,5]. Toxicological studies have suggested that peroxisome proliferation, hepatotoxicity, carcinogenicity, immunotoxicity, lipid metabolism, and developmental toxicity may be associated with chemical exposure to PFAS, especially PFOA [28,29]. The acute toxicity values are mostly >100 mg/L, which means that PFOA is not classified as a toxic substance according to the REACH Regulations (Appendix A). At low concentrations over long discharge periods, PFOA can cause sublethal effects, such as carbohydrate metabolism imbalance, osmoregulation, glucose content, histological anomalies at the level of the gonad, and thyroid and reproductive toxicity [30,31].

The aquatic environment is the most affected environmental compartment because these compounds, with endocrine disrupting properties, are released into the environment mainly through the treatment of municipal and industrial wastewater (treatment plants) [15]. Compared to most persistent organic pollutants (POPs) that tend to accumulate in adipose tissue, PFOA shows a strong affinity for protein-rich fish tissues (blood serum, liver, kidney, and gallbladder) [31]. At the laboratory level, the bioconcentration factor of PFOA was calculated for a species of carp, where values between 3.2–9.4, were recorded, indicating a high degree of bioconcentration [31]. The high stability of PFOA facilitates accumulation in aquatic organisms and makes the depuration stage long. It is considered that the main excretion route is the renal system, but there are studies that demonstrate that it can also be eliminated through the digestive and respiratory routes (at the level of the gills), but in very small quantities [32]. Therefore, PFOA enters an enterohepatic recirculation mechanism through which it is excreted from the liver in the bile to the small intestine and then reabsorbed and transported back to the liver [32]. In addition, PFOA can bind to the ovalbumin of embryos, facilitating maternal transfer and their defective development [32,33].

Laboratory studies on the bioconcentration and specific tissue distribution of PFAS in fish are essential for evaluating the degree of degradation of aquatic organisms and the effects on the environment. In 2015, a group of Swedish researchers [33] carried out a bioconcentration study on a zebrafish species, *Dario rerio* (males and females), continuously exposed to a concentration of 10 µg/L of isotopically labeled PFOA (^14^C-PFOA) for 40 days, followed by a depuration stage of 80 days. Compared to exposure concentration, bioconcentration values 100 times higher were reported in the liver and intestine, while in the brain, ovary, and muscle, the bioconcentration factors were 15–20 times higher. Through the antibiograms performed, it was confirmed that the largest amount of ^14^C-PFOA accumulated in the bile and intestines, which confirms the enterohepatic circulation, and, at the same time, the presence of ^14^C-PFOA was observed in the oocyte maturation process, having effects on embryonic development [33]. Recent studies aim to assess the bioaccumulation factors of PFAS in aquatic organisms and the associated ecological risk. In this sense, Gallocchio F. et al. [34] published a long-term monitoring study of over 100 fish specimens (*Brabus plebejus*—Italian barbel, European carp species, and catfish species—*Isilurus glanis*) from a contaminated area in the Veneto region (Italy). The contamination was previously attributed to a fluorochemical factory that mainly released perfluorooctanoic acid (PFOA), among other PFAS, into the environment and was detected in 79% of fish samples with an average concentration of 0.33 ug/Kg.

Freshwater fish represent a considerable and necessary food resource for the human diet. The chemical contamination of this resource not only represents a challenge for world organizations to detect and monitor but also find solutions to combat it. To the best of our knowledge, a single paper in the literature reported the bioconcentration of ^14^C-PFOA in zebrafish, on a laboratory scale, and the uptake and distribution of this organic pollutant in fish tissue [33]. A single paper reported the bioconcentration factor of PFOA in European carp, by evaluating PFOA bioaccumulation in the whole body of the fish twenty years ago [35]. European carp is a popular food fish in many parts of Europe. It is known that this species is especially efficient at adsorbing pollutants such as PFOA from their environment, so understanding and mitigating their exposure to this chemical is crucial. By understanding how this chemical accumulates in these fish, we can better assess the potential risk to human health. Additionally, this information can be used to develop strategies for reducing PFOA exposure, both for carp and humans. Ultimately, the bioconcentration of PFOA in carp is a critical issue that deserves our attention. By studying this chemical and its effects on fish and humans, we can work to create a safer and healthier environment. To fulfill the purpose of this study, a series of objectives were followed: (i) detection and quantification of target analytes as well as biodegradation products in fish organs (represented through common carp species) by LC-MS/MS; (ii) determination of the bioconcentration potential of PFOA in common carp; (iii) assessment of the potential risk for humans; (iv) identification and quantification of PFOA biotransformation products in fish organs; and (v) proposing a biotransformation pathway of PFOA at the level of common carp organs.

## 2. Materials and Methods

### 2.1. LC-MS/MS Methods Development

#### 2.1.1. Chemicals and Materials

Analytical standard PFOA (Perfluorooctanoic acid), isotopically labeled analog used as an internal standard (^8^C-PFOA), and metabolites, Perfluorobutanoic acid (PFBA), Perfluoropentanoic acid (PFPeA), Perfluorohexanoic acid (PFHxA), and Perfluoroheptanoic acid (PFHpA) were purchased from Sigma Aldrich (Darmstadt, Germany). Solvents used for the preparation of the mobile phase and extractions from water samples include methanol-Merck (Darmstadt, Germany), acetic acid, formic acid, and ammonium acetate Sigma-Aldrich (Taufkirchen, Germany). High-purity water was procured using the Millipore Milli-Q purification system acquired from Millipore (Darmstadt, Germany).

#### 2.1.2. Instrument Analysis

Chromatographic analysis of PFOA and its metabolites was performed using an Agilent 1260 liquid-chromatograph system (Agilent Technologies, Waldbronn, Germany), coupled to an Agilent 6410B triple-quadrupole mass spectrometer (Agilent Technologies, Waldbronn, Germany), equipped with an EIS source (electrospray ionization) working in negative mode (LC-MS/MS system). Chromatographic data were collected using Mass Hunter acquisition and control software version B.04.00 from Agilent Technologies, which was also used for data processing and quantification of compounds.

#### 2.1.3. Analytical Methods

Monitoring PFOA concentrations in the water samples was carried out by direct injection of 5 µL of water samples into a Zorbax Eclipse C18 column (2.1 × 100 mm, 3.5 μm), maintained at a temperature of 30 °C. PFOA was eluted from the chromatographic column, using Aq 5 mM ammonium acetate/MeOH (20/80 *v*/*v*) as a mobile phase. The mobile phase flow rate was 0.2 mL/min and was eluted in an isocratic manner. Multiple reaction monitoring was used as the MS/MS detection mode.

The determination of PFOA concentrations accumulated in fish organs and tissues, as well as the formed metabolites, was carried out using the same LC-MS/MS system. The separation of the compounds was achieved using the same Zorbax Eclipse C18 column (2.1 × 100 mm, 3.5 μm), thermostated at 30 °C. The mobile phase used consisted of Aq 5 mM ammonium acetate (A) and MeOH, in the gradient mode (Appendix A), at a 0.2 mL/min flow rate, using an injection volume of 20 µL aqueous extract. The EIS parameters were the following: drying gas temperature: 300 °C; drying gas flow rate: 8 L/min; nebulizer pressure: 50 psi; and voltage on the capillary: 2500 V. The elution of the two analytes was achieved in less than 3.5 min. The operational parameters of the mass spectrometer and the MRM chromatogram obtained for PFOA, and its transformation products are shown in Appendix A.

#### 2.1.4. Matrix Interference and Quantification

To estimate the consequences of the complex matrix of fish samples in the EIS source, the post-extraction addition method was chosen. The obtained extracts were contaminated with a known concentration of an analyte mixture and an internal standard and then analyzed. Matrix effects (% ME) were calculated as the percentage ratio between the analytical signal generated by the analyte in the sample (A_spiked matrix_) and the signal generated by the analyte in the standard solution (A_standard solution_), using the following equation:ME (%) = (A_spiked matrix_/A_standard solution)_ × 100(1)

The determined values (between 29–91%) for the target analytes generated a suppression of the analytical signal. The analytical recoveries were situated between 71–92%. LOQ values for PFOA in water samples and biological samples were 0.08 ng/g d.w., and for PFOA metabolites, it ranged between 0.02–0.06 ng/g d.w. All the measured concentrations of biological samples were background-corrected following the matrix effects and recoveries. All results are presented in Appendix A.

#### 2.1.5. Data Analysis

PFOA tends to bioconcentrate in fish tissues and organs, and the cited studies have shown a global spread [36]. To evaluate the bioconcentration level of PFOA in aquatic organisms, it is essential to calculate the bioconcentration factors (BCF). Bioconcentration represents the process by which a chemical substance enters an organism and/or is adsorbed through exposure to water within a certain time interval; it is a measure determined at the laboratory level. The bioconcentration factor (L/Kg) is estimated as a ratio between the concentration of a chemical compound in the body (C_biota_) and the concentration in water (C_water_) after the equilibrium state has been reached (Equation (2)).
BCF = C_biota_/C_water_(2)

A bioconcentration factor with a value greater than 1 indicates a lipophilic chemical compound. The value of the bioconcentration factors is proportional to the mass of the chemical compound adsorbed in the fish tissues and organs [37,38].

#### 2.1.6. Statistical Analysis

Statistical analysis was performed with Microsoft Excel 2017. For all experiments, the results were expressed as means (*n* = 3) ± standard deviation (SD). Correlations between PFOA concentration and its biotransformation products in fish organs and tissue were performed using Spearman’s correlation. The Spearman correlation coefficient is a statistic that ranges from −1.0 to +1.0. The closer the Spearman correlation coefficient is to ±1, the stronger the linear correlation is between the two variables. The criterion for significance was set at *p* < 0.05.

### 2.2. Conditions for Conducting Chronic Tests

#### 2.2.1. Biological Material

The biological material used in the toxicological experiments are specimens of European carp (*Cyprinus carpio*), widely known as common carp, weight 30–40 g/individual, and purchased in October 2021 from the NUCET Aquaculture Research and Development Station (according to Quality Certificate No. 738/21.10.2021). The specimens subjected to the bioconcentration study showed normal morphology, and no visible lesions were observed after acclimatization in laboratory conditions in the maintenance tanks of the aquatic biobase owned by the laboratory. Their feeding was carried out with specific artificial food for the purpose of survival and not growth. Food consisted of a complex based on grain, table plant extraction, yeasts, vitamins, and other minerals. The food ration represents 1% of the weight of the tested group and is recalculated after each stage of sampling and slaughter. Feeding was performed daily by hand. The food was Tetra Pond sticks from the trade that contain essential nutrients, trace elements, vitamins, and carotenoids.

The maintenance of the fish and the experiments were carried out in accordance with the Guide for the Care and Use of Laboratory Animals and according to the OECD recommendations regarding the reduction of the suffering of animals used in laboratory tests. The experimental studies were monitored by the Ethics and Professional Deontology Commission of INCD ECOIND (Ethical Opinion no. 4336/28.02.2022).

#### 2.2.2. Sample Preparation

During the exposure period, 3 fish dissection campaigns were organized (at 8 weeks, 12 weeks, and 14 weeks), and a fourth during the depuration stage, 3 weeks after the end of the exposure phase (in the manuscript, it is 17 weeks). For each level, 4 randomly selected specimens were sacrificed (16 specimens in total). The fish were sacrificed on the ice through spinal nerve dissection. The organs were weighed and stored in individual containers at 4 °C. The wet tissues (gills, skin, scales, muscle, liver, brain, gonad, intestine, kidneys, and gallbladder) were freeze-dried at −100 °C, crushed, and homogenized. The extraction procedure was performed using 20 mL glass ampoules, 250 mg of dry sample, and 10 mL of methanol. The mixture was vortexed for 1 min and sonicated for 30 min at 30 °C. After extraction, the solid mass was decanted at 3000 rpm (centrifuged for 10 min). The organic phase was collected, and the solid residue was submitted to a new extraction procedure. After completion, the organic layers were mixed and purified on silica gel. The mixture was evaporated almost to dryness in nitrogen flow, resuspended in 1 mL of water, and then moved to a vial. After maintaining at 4 °C, the precipitate was decanted after centrifugation at 14,000 rpm, for 10 min. The aqueous phase was subsequently moved to a 2 mL vial and investigated by LC-MS/MS.

#### 2.2.3. Long-Term Experimental Exposure

The experimental procedure aimed to evaluate the bioconcentration in fish and was carried out according to OECD Guideline 305-II: The minimized fish exposure test in an aqueous system (2012). The method of exposure in an aqueous system assumed the contamination of fish in an aqueous solution of PFOA and the estimation of the bioconcentration factor. To initiate the test, the fish specimens were measured and weighed to ensure homogenous batches, with 20 specimens each being selected for the test experiments (10 µg/L PFOA—Experiment 1 and 100 µg/L PFOA—Experiment 2) and another 20 specimens for the control experiment. The exposure concentrations were selected based on the literature concerning laboratory exposure studies on other fish species [33,34].

Initially (0 days), the selected specimens had the characteristics presented in Appendix A. The water used for the test was dechlorinated from mains water by aeration and was periodically analyzed, from a physical-chemical point of view, to meet the conditions imposed by the test method. No values above the permissible limits imposed by the test method were identified. The tests were performed in 100 L aquariums that allowed the preparation of 80 L of test solutions at a 20 °C ± 4 with constant aeration. The test was carried out in a semi-static system with dilution water every 48 h. Indicators such as: oxygen, pH, temperature, and conductivity were monitored daily before and after the renewal of the experimental solutions. The analyzed indicators respected the optimal test conditions recommended by the OECD, respectively: minimum oxygen of 4 mg/L, pH of 6–8.5, and temperature of 18–25 °C. The period of the bioaccumulation test was 17 w (14 w—exposure period and 3 w—depuration period). For all species, the characteristics of the sacrificed specimens are presented in Appendix A.

## 3. Results and Discussion

### 3.1. Dilution Water Characteristics

The physical-chemical parameters of the dilution water were periodically analyzed, with the water quality being in accordance with the OECD test conditions. Thus, in the exposure experiments, the average values of the physical-chemical parameters were: 7.71 ± 0.25 pH units (pH), 343 ± 33.7 µS/cm (conductivity), 7.37 ± 0.96 mgO_2_/L (COD), and 21.8 ± 0.85 °C (temperature). In the control experiment, the reported values were 7.90 ± 0.10 pH units (pH), 343 ± 17.6 µS/cm (conductivity), 8.33 ± 0.45 mgO_2_/L (COD), and 20.9 ± 0.87 °C (temperature).

### 3.2. Stability of PFOA in Water

To test the stability of PFOA in water, two experiments were performed. Under the same conditions as the exposure experiments, two aquariums without fish were contaminated with 10 µg/L PFOA and 100 µg/L PFOA. PFOA concentrations in both aquariums were analyzed daily for 96 h. It was observed that the measured concentrations of PFOA in water were close to the theoretical concentrations during the tested period. Thus, the average concentrations were 9.92 ± 0.02 µg/L for the nominal value of 10 µg/L and 9.98 ± 1.84 µg/L for the nominal value of 100 µg/L (Table 1).

### 3.3. PFOA Water Concentration

In Experiments 1 and 2, the measured concentrations of PFOA in water were close to the nominal concentrations (≥90%) throughout the exposure period. In Experiment 1, the mean PFOA concentrations during the exposure period were 9.03 ± 1.6 ug/L. In Experiment 2, measured PFOA concentrations over the exposure period were 92.1 ± 10.3 ug/L. PFOA was not detected in the control tanks in any of the experiments. No mortality was observed in either of the two experiments.

### 3.4. Test Fish Characteristics

At the end of the exposure period (14 w), the fish were visually analyzed, measured, and weighed. The initial data (0 days, Appendix A) were compared with the final data (Appendix A), and no modifications were observed. No mortalities or other visible physiological modifications were observed.

### 3.5. PFOA Occurrence in Tissue

The biological samples (gills, skin, scales, muscle, liver, brain, gonad, intestine, kidneys, and gallbladder) collected during both exposure experiments (10 ug/L and 100 ug/L PFOA) were subjected to extraction and analyzed in order to determine the concentrations of PFOA present in the organs at certain time intervals (8 w, 12 w, 14 w, and 17 w). The concentration values of PFOA in each organ are given in Appendix A for Experiment 1 and in Appendix A for Experiment 2.

In Experiment 1, the highest concentrations were determined in the gallbladder, with the PFOA level being up to 2570 ng/g d.w. The PFOA values observed in organs at each slaughter campaign varied differently, but the lowest concentrations were determined after the purification period (Figure 1). In the intestine and gonad, the PFOA values found after 8 weeks from the beginning of the experiment were 97.8 and 150 ng.g d.w., respectively, with the values increasing until the second sampling campaign (12 w), after which a decrease in the concentration was observed within the 14 w, and significantly lower values after the purification period—17 w (Figure 1a). The difference between the concentration levels determined between the weeks 12 and 14 may suggest the biotransformation of PFOA in these organs. In the case of PFOA determined in the kidneys and brain, relatively constant values were determined after 8 w of exposure (up to 435 ng/g d.w. in the kidneys and up to 275 ng/g d.w. in the brain), after which the concentration level started to decrease, reaching up to 265 ng/g d.w. in the kidneys and up to 58 ng/g d.w. in the brain (Figure 1b). In the liver and gills, the highest PFOA values were determined in the samples collected at 8 w, with up to 543 ng/g d.w. in the liver and up to 426 ng/g d.w. in the gills. PFOA levels decreased similarly in the case of both organs after 14 w of exposure, reaching up to 38 ng/g d.w. in the liver and up to 90 ng/g d.w. in the gills, after the purification period (Figure 1c). The continuous decrease of PFOA concentration levels in both organs indicates the metabolization of PFOA.

The highest PFOA values were observed in the gallbladder, with maximum values recorded at 8 weeks of exposure, and, in this case, a continuous decrease in PFOA concentration was observed throughout the test (Figure 1c). In the muscles, the concentration of PFOA increased between weeks 8 and 12 of exposure, subsequently remaining constant, and suddenly decreasing during the purification period (Figure 1d). A high concentration of PFOA was determined in the scales and skin in samples collected after 8 w of exposure (up to 286 ng/g d.w. and 318 ng/g d.w.), and this increased throughout the following two sampling campaigns (12 w and 14 w) and decreased to values of up to 3.12 ng/g d.w. and 4.87 ng/g d.w. after the purification period (17 w, Figure 1d).

Considering the average of the PFOA values determined in the organs during the exposure period, in which the biological specimens were kept in contact with the dilution water contaminated with PFOA, the ascending order of the PFOA determined in the fish organs was as follows: gonad < intestine < liver < brain < gills < muscle < scales < skin < kidney < gallbladder. The higher concentration determined in the gills compared to the concentration determined in the intestine indicates that the main way of PFOA penetration into the studied aquatic organism is not ingestion but the breathing function, namely through the gills.

By evaluating the amount of PFOA identified in the organs (as a sum), we can state that the highest values were determined after 8 w of exposure (up to 5153 ng/g d.w.) and continuously decreased, with the lower values being determined during the purification period, up to 887 ng/g d.w. (Appendix A). Evaluating the percentage distribution of PFOA concentration in the organs, it can be stated that the highest PFOA values were determined in the gallbladder and kidneys in all four slaughter campaigns (Appendix A). The increase in the percentage values of PFOA at the level of the kidneys throughout the test may indicate that the main way of elimination of PFOA left unchanged is the urinary tract [38].

In Experiment 2, the PFOA concentration levels determined in the organs were significantly higher compared to those observed in Experiment 1 (Appendix A). The highest values of PFOA were identified in the gallbladder and kidneys. After 8 w of exposure, the concentration of PFOA was up to 18,640 ng/g d.w. in the gallbladder and up to 13,581 ng/g d.w. in the kidneys. Later, these concentrations registered a rapid decrease throughout the test, reaching up to 1026 ng/g d.w. in the gallbladder and up to 179 ng/g d.w. in the kidneys in the samples analyzed after 14 w of exposure. During the purification period, the concentration of PFOA continued to decrease, although much more slowly (Figure 2a).

In the gonad, brain, gills, and scales, the concentration levels of PFOA were between 1758 and 4222 ng/g d.w. after 8 w exposure. The determined values decreased throughout the test period, with the PFOA concentration being situated between 78.5 and 187 ng/g d.w. in the samples collected after 14 w of exposure and between 15.5 and 57.5 ng/g d.w. after the purification period (Figure 2b). In muscle and intestine, the PFOA values determined after 8 w of exposure were 981 and 828 ng/g d.w., increasing to 1565 ng/g d.w. in muscle and 1136 ng/g d.w. in the intestine in the samples analyzed after 12 w of exposure. The PFOA values determined in muscles and intestines after 14 w of exposure followed an accelerated decrease, continuing even after the purification period, although much slower (Figure 2c). In the liver and skin, the highest PFOA levels were determined after 8 w of exposure, with concentrations determined up to 2729 ng/g d.w. and 2526 ng/g s.u; The values decreased throughout the testing and purification period, reaching 42.4 ng/g s.u in the liver and 23.1 ng/g s.u in the skin (Figure 2d).

Evaluating the average of PFOA values determined in organs throughout the tested period, the ascending order of PFOA values determined in organs was the following: intestine < muscle < brain < scales < skin < liver < gonad < gills < kidney < gallbladder. Similar to Experiment 1, in Experiment 2, the higher concentration determined in the gills compared to that determined in the intestine indicates that the main way PFOA penetrates into the studied aquatic organism is not through ingestion, but through the breathing function, namely the gills. By analyzing the amount of PFOA identified in the organs (as a sum), we can state that the highest values were determined in the first slaughter campaign (up to 49,500 ng/g d.w.), and it continuously decreased in the following campaigns, with the lower values being determined during the purification period, up to 756 ng/g d.w. PFOA (Appendix A). Evaluating the percentage distribution at the level of the organs, it can be stated that the highest PFOA values were determined in the gallbladder and kidneys, in all four slaughter campaigns (Appendix A). The increase in the percentage values of PFOA at the level of the kidneys throughout the test may indicate that the main way of elimination of PFOA that has not been metabolized is excretion.

The distribution pattern and the evolution of the PFOA concentration in the fish organs proved to be similar, both at low test concentrations (10 µg/L) and at high concentrations of PFOA (100 µg/L). The reason why PFOA concentrations are high in the gallbladder and liver is probably due to the enterohepatic circulation of this chemical. This mechanism has also been observed in laboratory studies with rainbow trout [39] and rodents [40,41]. Enterohepatic recirculation increases the risk of hepatotoxicity, and, in fact, most PFASs show similar toxicity profiles, with the liver being the main target organ [33,34]. Moreover, the accumulation of PFOA in brain tissues may represent a risk, acting as a potential neurotoxic agent for development [34]. The obtained data are also supported by the changes that occurred in the hepatosomatic (HIS) and gonadosomatic (IGS) indices as a result of exposure to PFOA, which confirm the endocrine effect of this compound, especially in the case of 100 µg/L after 14 w, with the effect reducing in the depuration phase (Figure 3).

Many papers focused on the in situ determination of PFOA in fish, and very few evaluated the in vivo assimilation of PFOA in fish organs [33,42,43,44,45]. Ulhaq et al. evaluated the assimilation, distribution, and elimination of PFOA in zebrafish, under conditions similar to the present study. After 40 days of exposure to 10 ug/L PFOA, the concentrations determined in fish fell between 489 and 239 ng/g in males and females [33]. Compared to this study, the values determined in the present experiment are higher, but a comparison cannot be made due to the differences between the fish species tested. Compared to in situ studies, the PFOA values determined in the present study are higher than those published in the literature [33,46]. However, this discrepancy is normal considering the use of a PFOA test concentration higher than the level present in surface waters and keeping it constant during the exposure periods in order to observe in a short period of time the bioaccumulation and biotransformation of these chemicals in fish organs.

### 3.6. Bioconcentration Factors of PFOA

The bioconcentration factors (BCF) were calculated based on the PFOA concentrations quantified in the organs and the average of the PFOA concentrations obtained over the 8, 12, 14, and 17 exposure weeks in water. A value of BCF > 1 indicates bioaccumulation in fish organs, while a BCF < 1 indicates no bioaccumulation [47]. PFOA was detected in all analyzed organs. The BCF values vary depending on the target organ, with the greatest values in the gallbladder and kidney (Table 2).

The first 8 weeks of exposure showed PFOA in the range of 6.78 to 233, with a potential for bioaccumulation of >1 in all organs. A high BCF in gallbladder, kidneys, and liver suggests, with more time, the hypotheses of enterohepatic circulation. After 12 weeks, the bioaccumulation of PFOA (10.1 to 164) was recorded for all organs analyzed in the following estimated order: gallbladder, > kidneys > muscle > skin > scales > brain, > intestine > gonads > gills > liver. After 14 weeks, the PFOA BCFs decreased in all organs. The equilibrium phase was obtained only in the case of certain target organs (scales, skin, muscles, and kidneys) at a concentration of 10 µg/L between 12 w and 14 w. In the case of the other organs, no equilibrium phase is observed, and, at 14 w, the BCF values are much lower, indicating a reduced bioaccumulation of PFOA which is probably metabolized and is accumulated or eliminated in other forms, or an adaptation to the toxic process occurred. During the experiment, PFOA recorded an average organ-specific BCF of 8.4 to 158, indicating a high potential for bioaccumulation in fish tissues. In the depuration phase (17 w), the BCF values decreased proportionally with the PFOA concentration removed. The BCF values obtained were confirmed by specialized literature, respectively 0.52 to 0.44 for crucian carp [48] and 5.1 to 9.4 for common carp [49]. According to the NITE-CHRIP database on risk assessments and laws and regulations of the PFOA [50], the following BCF was registered: 4.2 and 9.4 for the whole body in a 28 day test on common carp.

### 3.7. Potential Human Risk Assessment

The risk to human health from fish consumption was estimated by calculating the average daily intake (ADI) of PFOA. The study on daily fish consumption in Romania was not carried out in this paper. According to the European Commission, the average person living in the EU consumes 24 kg (live weight) of fish or seafood per year, while in Romania, fish consumption was estimated at 8.11 kg per year, meaning 0.37 g/Kg body weight (bw)/day (supposing a bw of 60 kg) [51]. Based on this value, the ADI was estimated by multiplying the PFOA measured concentrations in muscle (ng/g dw; this being considered the most consumed part of the fish) by the average fish consumption (ng/kg/day dw) (Equation (3)).
ADI = Conc_PFOA_ × Fish consumption(3)

According to the European Food Safety Authority (EFSA), the tolerable weekly doses (TWI) have been evaluated as a sum of PFOS, PFOA, PFHxS, and PFOS at 4.4 ng/kg bw per week (meaning 0.63 ng/kg bw/day) [52]. The average concentration of PFOA determined in the muscles in the three sampling campaigns for both experiments was used to estimate the ADI value. The estimated ADIs were much higher than the ADIs established by EFSA, being 99 ng/Kg/day in Experiment 1 and 320 ng/Kg/day in Experiment 2. Using the obtained values for ADIs, the human risk (HR) was calculated as a ratio between the average value and the reference value. HR > 1 suggests that the exposure level exceeds the reference concentration.

The HR values determined for PFOA in muscle tissue were orders of magnitude above 1 (157 and 508 in Experiment 1 and Experiment 2, respectively), indicating that the level of PFOA in muscle poses a high risk to human health. However, in this study, the HR values are empirical, based on bioconcentration studies performed in laboratory conditions. These conditions do not reflect the real conditions, being achieved using very high concentrations of PFOA and a very short exposure time, unlike the real situation, where in aquatic ecosystems the concentrations of PFOA found in surface waters are of the order of ng/L but the exposure period is a long one (up to several years). In addition, in real ecosystems, biomagnification can increase the risk to human health due to the consumption of fish located higher in the food chain.

### 3.8. PFOA Biotransformation Product Identification in Fish Organs

In the biological samples taken during the two Experiments (1 and 2), four biotransformation products of PFOA were identified, namely PFBA, PFPeA, PFHxA, and PFHpA (Appendix A). The concentration levels of the biotransformation products varied between <LOQ and 120 ng/g d.w. in Experiment 1 and between <LOQ and 204 ng/g d.w. in Experiment 2. In most organs, and in both experiments, the sum of the values determined for the biotransformation products of PFOA proved to be higher in the second slaughter campaign, with the highest values being obtained in the kidney, brain, gonad, liver, and gallbladder in both tests (Figure 4). An exception to this pattern was observed only in the case of scales, where the highest concentrations of biotransformation products were determined after 14 weeks of exposure. After this period, in the case of the other organs, the amount of biotransformation products decreased by at least half compared to the previous campaign, with the only exception being the gonad, where the amount of biotransformation products remained constant, while after the period of purification, the values determined in the organs decreased considerably in both tests. In all biological samples, the dominant biotransformation product was PFBA, and the highest concentrations were determined in the kidney (97.5/204 ng/g d.w.), brain (104/200 ng/g d.w.), gonad (42.0/62.0 ng/g d.w.), liver (58.5/80.6 ng/g d.w.), and gallbladder (120/176 ng/g d.w.) Values were obtained after 8 weeks of exposure in both tests.

The biological samples analyzed after 12 and 14 weeks of exposure and after the purification period (17 w) were dominated by PFBA, the majority biotransformation product of PFOA. It was observed that, in most cases, the biodegradation products identified in the organs taken after 8 weeks of exposure were those with a longer aliphatic chain, namely PFHpA, PFHxA, and PFPeA, with their presence identified especially in the scales (0.03 ng/g d.w PFHpA and 0.03 ng/g d.w PFPeA), kidneys (0.11 ng/g d.w PFHpA and 0.32 ng/g d.w PFPeA), brain (0.03 ng/g d.w PFHpA), gallbladder (0.37 ng/g d.w PFHpA, 0.45 ng/g d.w PFHxA, and 0.12 ng/g d.w PFPeA), and gonad (0.07 ng/g d.w PFHpA) in Experiment 1 (Appendix A and Figure 5a), as well as in the scales (0.22 ng/g d.w PFHpA, 0.06 ng/g d.w PFHxA, and 0.03 ng/g d.w PFPeA), kidney (1.89 ng/g d.w PFHpA, 0.99 ng/g d.w PFHxA, and 0.32 ng/g d.w PFPeA), brain (0.51 ng/g d.w PFHpA), liver (0.13 ng/g d.w PFHpA), intestine (0.09 ng/g d.w PFHpA and 0.12 ng/g d.w PFPeA), and gallbladder (3.79 ng/g d.w PFHpA, 0.90 ng/g d.w PFHxA, and 0.15 ng/g d.w PFPeA) in Experiment 2 (Appendix A and Figure 5b).

The highest level of biotransformation products was in the liver, kidney, brain, and gallbladder, while the lowest was observed in the scales, skin, muscle, and gills, a finding that is consistent with previous studies [53]. The liver is an organ rich in proteins, recognized for its function of metabolizing various organic compounds. Thus, a variety of organic pollutants from the environment tend to accumulate, especially in fish liver [54]. The kidney is the main excretory organ, as its function is the elimination of secondary products resulting from metabolism in the body [54]. Thus, the concentrations of organic pollutants found in the environment are generally much higher in the liver and kidneys than in muscles and other tissues. Published papers also reported that perfluoroalkyl compounds have a strong affinity for liver proteins, as perfluoroalkyl compounds have a greater tendency to accumulate in the liver than in the kidneys [55,56]. Contrary to this observation, for certain perfluoroalkyl compounds, their concentrations in the fish kidney proved to be higher than those in the liver; this situation is also observed in the present study. A hypothesis is that long-chain perfluoroalkyl compounds resemble saturated fatty acids and are able to be transported from the liver, resulting in lower hepatic levels of long-chain compounds [57]. Another explanation is that long-chain perfluoroalkyl compounds are not filtered in the kidney at the same rate as shorter-chain compounds, leading to a preferential accumulation of long-chain perfluoroalkyl compounds in the kidney [40].

### 3.9. Spearman Correlations between PFOA and Biotransformation Products in Fish Organs

The relationship between PFOA and its biotransformation products identified at the level of the organs was analyzed using Spearman correlations.

In Experiment 1, strong negative correlations were observed in the gills, between PFOA and PFHpA (r = −0.873; *p* = 0.16), and positive correlations between PFHpA and PFHxA (r = 0.738; *p* = 0.042), PFHpA, and PFPeA (r = 1.000; *p* = 0.000), but also between PFPeA and PFBA (r = 0.862; *p* = 0.033), suggesting the direct biotransformation of PFOA in PFHpA, followed by the transformation of the latter one into compounds with a shorter aliphatic chain, PFHxA, and PFPeA, and obtaining the biotransformation product PFBA through PFPeA metabolization (Appendix A). In Experiment 2, there was a strong correlation between PFOA and PFHpA (r = −0.877; *p* = 0.037) and between PFOA and PFPeA (r = −0.927; *p* = 0.020) observed in the gills, suggesting the direct biotransformation of PFOA in compounds with shorter aliphatic chains, namely PFHpA and PFPeA. PFPeA concentration values determined in gills were correlated with PFHpA (r = 1.000; *p* = 0.000) and PFHxA (r = 0.894; *p* = 0.043), indicating a common source of formation, but also between PFPeA and PFBA (r = −0.898; *p* = 0.031), hinting that the major biotransformation product was obtained from PFPeA metabolization (Appendix A). A similar pattern of correlations between the target compounds was observed in the kidneys in both experiments. Thus, strong correlations were observed between PFOA and PFHpA (r = 0.91; *p* = 0.033/r = −1.000; *p* = 0.000), suggesting the metabolism of PFOA in its analog with seven carbon atoms in the aliphatic chain (Appendix A). Close relationships were also observed between pairs of compounds in which the length of the aliphatic chain differs by one carbon atom, namely between PFHpA and PFHxA (r = 0.775; *p* = 0.025/r = 1.000; *p* = 0.000), between PFHxA and PFPeA (r = 0.747; *p* = 0.050/r = 1.000; *p* = 0.000), as well as between PFPeA and PFBA (r = 0.813; *p* = 0.047/r = 0.846/*p* = 0.039). This information leads to the fact that, in the kidney, the metabolism of PFOA takes place successively, up to PFBA. Additionally, in Experiment 2, a strong positive correlation was also calculated for PFOA and PFHxA, and for PFOA and PFPeA (r = 1.000, *p* = 0.000), suggesting that PFOA is the parent compound of both PFHxA and PFPeA compounds. In the brain, strong correlations were observed between PFOA and PFHpA (r = 1.000, *p* = 0.002/r = −0.869, *p* = 0.027), indicating that PFHpA is the main metabolite of PFOA (Appendix A). Strong correlations were determined between PFHpA and PFHxA in both experiments (r = 1.000, *p* = 0.000), between PFHpA and PFPeA in Experiment 2 (r = 0.832, *p* = 0.036), but also between PFHxA and PFPeA (r = 0.899, *p* = 0.023/r = 0.932, *p* = 0.033), indicating that PFHxA was obtained from PFHpA metabolization, while PFPeA was obtained from both PFHpA and PFHxA metabolization. In addition, the PFBA concentration value determined in Experiment 2 was highly correlated with PFPeA values, indicating that PFBA was obtained from a PFPeA biotransformation.

In both experiments, strong negative correlations were observed in the gonads between PFOA and the biotransformation products PFHpA (r = −0.858, *p* = 0.042/r = −0.872, *p* = 0.036) and PFHxA (r = −0.949, *p* = 0.041/r = −0.833, *p* = 0.038), indicating that PFOA is directly responsible for both metabolite formations (Appendix A). A strong positive correlation was also calculated for PFPeA and PFHpA (r = 0.882, *p* = 0.033/r = 0.833, *p* = 0.037) and PFHxA (r = 0.792, *p* = 0.032/r = 0.832, *p* = 0.036), respectively, suggesting PFPeA as the main biotransformation product from both analog compounds with seven and six carbon atoms in the aliphatic chain. PFBA content was correlated with PFPeA in both experiments (r = 0.822, *p* = 0.041/r = 0.895, *p* = 0.011), suggesting that PFPeA is the parent compound of the PFBA product. In the liver, negative correlations were determined between PFOA and PFHpA (r = −0.823; r = 0.020/r = −1.000; *p* = 0.000) and between PFOA and PFPeA, in both experiments (r = −0.839, *p* = 0.045/r = −0.889, *p* = 0.032), indicating that PFHpA and PFPeA were obtained directly from the PFOA metabolism process (Appendix A). Very strong correlations were determined between PFPeA and PFHpA (r = 1.000, *p* = 0.00), and between PFPeA and PFHxA (r = 0.926, *p* = 0.041/r = 0.883, *p* = 0.048) but also between PFBA and PFPeA (r = 0.822 *p* = 0.043/r = 0.807 *p* = 0.049), suggesting that PFPeA is the main biotransformation product of PFHpA and PFHxA, but also that the PFBA biotransformation product was obtained from PFPeA metabolization. In the gallbladder, a similar pattern was observed for both experiments. Strong relationships were determined for PFOA and PFHpA (r = 0.855, *p* = 0.044/r = 0.861, *p* = 0.048), PFHpA, and PFHxA (r = 1.000, *p* = 0.000), PFHxA, and PFPeA (r = −0.792, *p* = 0.048/r = −0.804, *p* = 0.033) and between PFPeA and PFBA (r = 0.783, *p* = 0.039/r = 0.897, *p* = 0.020), which indicates that the metabolic processes occur successively through the loss of a −CF_2_ group, up to the biotransformation product with four carbon atoms in the aliphatic chain (Appendix A). PFOA concentration levels determined in the intestine correlated with its counterpart with one less −CF_2_ group, PFHpA (r = 0.789, *p* = 0.011/r = −0.847, *p* = 0.029), and between PFHpA and PFHxA (r = 1.000, *p* = 0.000/r = 0.949, *p* = 0.033), indicating a successive metabolization process of PFOA (Appendix A). Moreover, strong relationships were observed between PFPeA and PFOA (r = 1.000, *p* = 0.000/r = 0.930, *p* = 0.024), PFHpA (r = 0.848, *p* = 0.20/r = 1.000, *p* = 0.033), and PFHxA (r = 0.949, *p* = 0.020/r = 0.949, *p* = 0.031), suggesting that PFPeA is directly obtained from compounds with longer aliphatic chains. In muscle and skin, Spearman’s correlation test showed a high correlation between PFOA and PFHpA, PFHpA and PFHxA, PFHxA and PFPeA, and also between PFPeA and PFHpA, only in the skin (*p* = 0.05) (Appendix A). Strong positive correlations were observed between PFOA and PFHpA, PFOA and PFHxA, PFHpA and its homologues with six, five, and four carbon atoms in the aliphatic chains, but also between PFBA and PFPeA and PFHxA (*p* = 0.05) in scales (Appendix A). These results may indicate a common source of origin, namely probably the dilution water from aquariums, in which both PFOA and its biotransformation products excreted during the exposure and purification periods are present.

### 3.10. PFOA Biotransformation Pathway in Fish Organs

In Experiment 1, the highest concentrations of both PFOA and biotransformation products were determined in the gallbladder, kidney, brain, liver, and gonad (Appendix A). In the gallbladder, the decrease in PFOA concentration could be correlated with the formation of biotransformation products. After the first 8 w of exposure, the presence of long aliphatic chain products was identified, namely PFPeA, PFHpA, and PFHxA. In the samples analyzed after 12 w of exposure, an increase in the concentration of the three products was observed, up to 1 ng/g d.w., but also the formation of the PFBA product at a much higher level, exceeding 120 ng/g d.w. After 14 w of exposure, along with the decrease in the concentration of PFOA, the increase in the concentration of the three products with a long aliphatic chain is still noticeable. At the same time, for PFBA, its decrease can be observed up to half compared to the previous campaign, indicating a metabolism of it at the level of the gallbladder into compounds that were not the subject of this study. After the purification period, the only biodegradation products identified in the gallbladder were PFPeA and PFBA. In the kidneys, the highest concentration of PFOA was identified after 8 w of exposure (up to 435 ng/g d.w.), after which it decreased constantly, reaching 265 ng/g d.w., and in the kidneys, the formation of long aliphatic chain products was observed from the first slaughter, their concentration level increasing throughout the test but not exceeding 3 ng/g d.w. Unlike them, the presence of PFBA was identified starting with 12 w of exposure in concentrations up to 100 ng/g d.w., this value decreasing until the end of the test period to 13 ng/g d.w. In the brain, between the first two sampling campaigns (8 w and 12 w), PFOA shows a slight increase, after which its concentration decreases until the end of the test period. After 12 w of exposure, the formation of a rather high concentration of PFBA is observed. The decrease in PFOA concentration after 14 w of exposure coincides with the formation of biotransformation products with five, six, and seven carbon atoms in the aliphatic chain. In the gonad, the concentration of PFOA increases in the samples taken in the first two campaigns, after which it decreases. The decrease in PFOA concentration coincides with the formation of biotransformation products with long aliphatic chains, with PFHxA being the majority, as its concentration even increased during the purification period. The formation of PFBA can be observed from the second sampling campaign, with its level remaining almost constant during the experiment, but decreasing during the purification period.

In the liver, the highest value of PFOA was determined in the samples taken after 8 w of exposure, after which it fell rapidly until 12 w of exposure and then more slowly until the third sampling campaign, continuing to slowly decrease during the purification period. The rapid decrease in the concentration of PFOA observed in the second sampling campaign (12 w) was accompanied by the formation of two biotransformation products in significant quantities, namely PFBA and PFHxA, up to 58 and 37 ng/g d.w., after which the concentrations of these two products decreased. In addition, in sampling campaign 2, the formation of PFPeA and PFHpA is also observed, and the values of these two compounds increase during the experiment, including during the purification period. In the gills, the highest value of PFOA was determined after the first 8 w of exposure, also indicating the main way through which PFOA enters the fish body; at the same time, a significant amount of PFBA was also observed. As the concentration of PFOA decreases after 12 w of exposure, the value of PFBA increases, suggesting the rapid metabolism of PFOA in PFBA at the level of the gills and the formation of much smaller quantities of PFPeA, PHHxA, and PFPeA. In the next sampling campaign, the concentrations of the four biotransformation products decreased, continuing even after the purification period for PFHxA and PFHpA. The same thing happened with PFOA while the concentration of PFBA increased, thus indicating the transformation of the two products with a larger aliphatic chain and of PFOA in the product with the smallest aliphatic chain studied, namely PFBA. In the intestine, the amount of PFOA continues to accumulate during the first 12 w of exposure, after which it decreases until the end of the experiment, including the purification period. The presence of PFBA is noted after 12 w of exposure, being the highest value recorded at the level of this tissue, and its concentration subsequently decreasing during the experiment. In skin, muscles, and scales, PFBA was the majority biotransformation product, with the concentration level of products with longer aliphatic chains being extremely low.

Similar to Experiment 1, and in the case of Experiment 2, both the presence of PFOA and its biotransformation products were identified in all organs, but in much larger quantities. The highest concentrations were recorded in the gallbladder, kidney, brain, liver, and gonad (Appendix A). The highest amounts of PFOA were determined in the gallbladder and kidneys. At the level of both organs, PFOA was determined in high quantities after 8 w of exposure, with up to 18,600 ng/g d.w. in the gallbladder, and up to 13,500 ng/g d.w. in the kidneys. This behavior was maintained in the gallbladder, kidney, and liver. Thus, although the highest concentrations of PFOA were identified after 8 w of exposure, they decreased similarly throughout the experiment. The decrease in PFOA concentration observed after 12 w of exposure coincided with a sharp increase in the biotransformation product PFBA, whose value subsequently decreased in sampling campaign 3 (14 w of exposure) and continued to decrease during the period of purification. The same cannot be said about the other biotransformation products, PFPeA, PFHxA, and PFHpA. Their presence was observed from the first sampling campaign, and their concentrations continued to increase throughout the experiment, including during the purification period. However, the concentration levels of the biotransformation products with a higher aliphatic chain were much lower compared to PFBA. However, their presence and behavior indicate that, although PFOA is very quickly biotransformed into PFBA, the intermediates PFPeA, PFHxA, and PFHpA are observed, and their concentrations increase during the experiment, but the process of passing from a product with a longer aliphatic chain to one with a shorter aliphatic chain is so fast that those with a long aliphatic chain do not have time to accumulate in large quantities in the organs. In the brain and gonad, the maximum amount of PFOA was determined in the first sampling campaign, after which it decreased throughout the experiment and during the purification period. In the case of these two organs, PFBA also proved to be the majority biotransformation product, with the highest values determined in the samples obtained after the second sampling campaign, after which its concentration decreased, remaining somewhere in the tens of ng/g d.w. after the purification period. Regarding the other three products, their formation was observed in the first sampling campaign at an extremely low level, but their values increased, obtaining a maximum of PFPeA, PFHxA, and PFHpA in the third sampling campaign.

In the gills, compared to the intestine, the concentration of PFOA was approximately four times higher in the samples taken after 8 w of exposure, indicating that the main way through which PFOA enters the fish body is through to the breathing function. The majority product, PFBA, is also identified from the first samples, its value registering an increase in 12 w of exposure, after which it decreases until the end of the experiment, with a slight increase during the purification period. In the case of the intestine, the formation of PFBA is observed from the second sampling campaign (12 w of exposure), when we also talk about the maximum value determined in this organ (7.6 ng/g d.w., a much lower value compared to those determined in the other organs), after which the concentration decreases until the end of the experiment. The same pattern of the biotransformation products is observed in the intestine, skin, and muscles. Biodegradation products with four, five, and six carbon atoms in the aliphatic chain were identified from the first sampling campaign, with their concentration increasing slightly throughout the experiment and a slight decrease after the purification period.

As a conclusion of this section, with the decrease in PFOA concentration, the initial formation of biotransformation products with long aliphatic chains, namely PFPeA, PFHxA, and PFHpA, was observed, and the increase in concentrations along the three slaughter campaigns, but up to an extremely low level compared to the PFBA biotransformation product. In all organs, the highest PFBA values were determined after 12 w of exposure, after which the PFBA level decreased in the following two campaigns, as a result of its biotransformation into compounds that are not the subject of this study. After the purification period, the amount of PFOA present in the organs was shown to extremely decrease, but not definitively, while the biotransformation products could still be identified even after this period. PFBA proved to be the majority biotransformation product in all organs, and the concentration level identified for the other products was extremely low by comparison. There are organs in which, in certain slaughter campaigns, the concentration could not be quantified. In most cases, the process is fast, and the biotransformation products with a longer aliphatic chain do not have time to bioconcentrate in the organs. Thus, we propose the biodegradation scheme according to Figure 6.

The biotransformation process of PFOA into PFBA as the majority product in organs is particularly important information, especially due to the fact that neurotoxicity is directly proportional to the increase in the length of the carbon chain [58]. For example, in a PFAS exposure study in zebrafish embryos/embryonic cells, sulfonic acid toxicity was greater than carboxylic acid, with a mortality rate that increased with carbon chain length (PFOA > PFHxA > PFBA) [59]. Thus, the presence of PFBA in higher concentrations compared to PFOA values in fish organs represents a lower risk for fish health.

## 4. Conclusions

The present study evaluated the PFOA concentration in *Cyprinus carpio* and the distribution pattern of the biotransformation products in the organs, in a long-term laboratory test exposure, at 2 concentrations (10 and 100 µg/L PFOA). PFOA accumulated in all organs, with the ascending order of the average concentration values being the following: gonad (136 ng/g d.w.) < intestine < liver < brain < gills < muscle < scales < skin < kidney < gallbladder (1668 mg/g d.w.) in Experiment 1, and intestine (686 ng/g d.w.) < muscle < brain < scales < skin < liver < gonad < gills < kidney < gallbladder (9474 ng/g d.w.) in Experiment 2. In both experiments, a higher concentration was observed in the gills compared to the one determined in the intestine, indicating that the main route of PFOA penetration into the studied aquatic organism is the breathing function. PFOA bioaccumulation in the gallbladder and liver is due to enterohepatic circulation. Enterohepatic recirculation can increase the risk of hepatotoxicity. In addition, the accumulation of PFOA in the brain may also represent a risk, acting as a potential neurotoxic agent for development. The average BCF values were >1 for all organs, being between 5.97 and 158 L/kg, signifying a high bioconcentration potential in all organs. Moreover, the values calculated in Experiment 1 proved to be higher than those determined in Experiment 2. Four biotransformation products of PFOA were identified in the biological samples, namely: PFBA, PFPeA, PFHxA, and PFHpA. In most organs, the sum of the values determined for the biotransformation products of PFOA proved to be higher in the second slaughter campaign, with the highest values being obtained in the kidney, brain, gonad, liver, and gallbladder in both experiments. In the case of both experiments, PFBA proved to be the major biotransformation product in all organs, with the concentration levels identified for the other products being extremely low by comparison. We can conclude that the biotransformation process of PFOA is fast, the biotransformation products with a high aliphatic chain (four, five, and six carbon atoms) are quickly metabolized into PFBA and do not accumulate in the fish organs in high concentrations.

The results obtained in this study indicate that the exposure of one of the most consumed fish species in Europe (*Cyprinus carpio*) to PFOA could present greater risks than expected.

## Figures and Tables

**Figure 1 foods-12-01423-f001:**
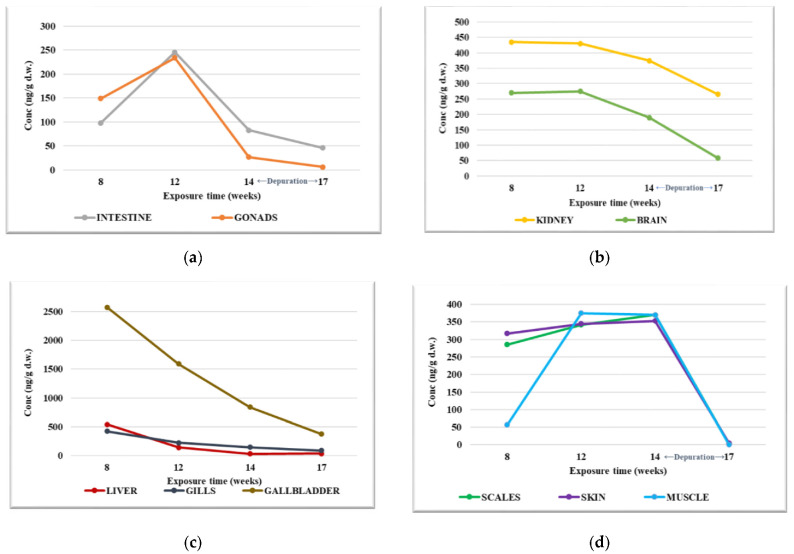
PFOA concentration levels determined in fish organs during the bioconcentration in Experiment 1: (**a**) intestine and gonads; (**b**) kidney and brain; (**c**) liver, gills, gallbladder; (**d**) scales, skin, muscle.

**Figure 2 foods-12-01423-f002:**
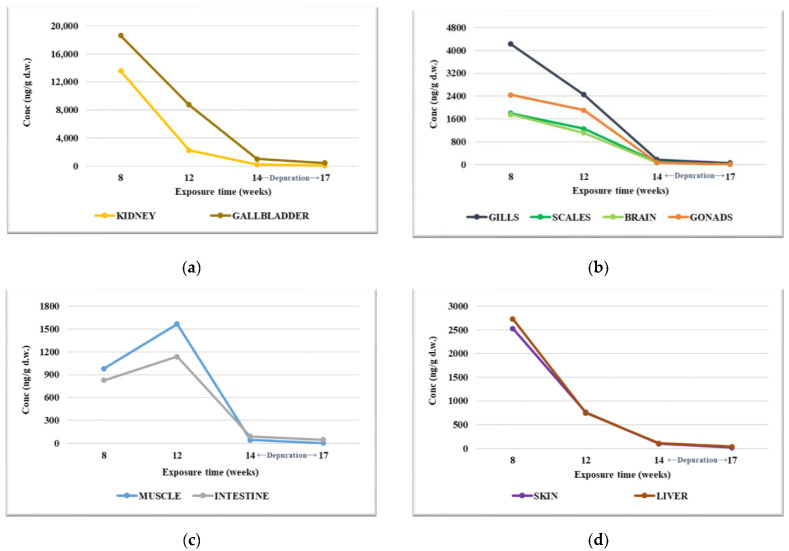
PFOA concentration levels determined in fish organs during the bioconcentration in Experiment 2: (**a**) kidney and gallbladder; (**b**) gills, scales, brain and gonads; (**c**) muscle and intestine; (**d**) skin and liver.

**Figure 3 foods-12-01423-f003:**
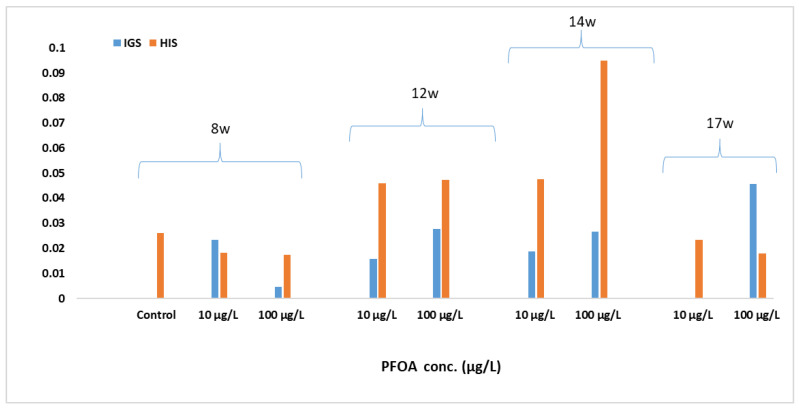
Hepatosomatic (HIS) and gonadosomatic (IGS) indices under the action of PFOA at 10 and 100 µg/L exposures at 8 w, 12 w, 14 w, and 17 w.

**Figure 4 foods-12-01423-f004:**
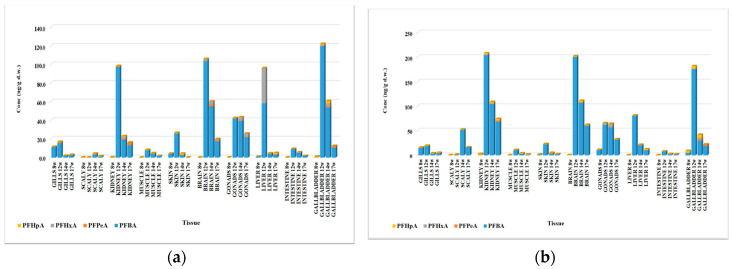
Variation of identified metabolite concentrations in fish organs in Experiment 1 (**a**) and Experiment 2 (**b**).

**Figure 5 foods-12-01423-f005:**
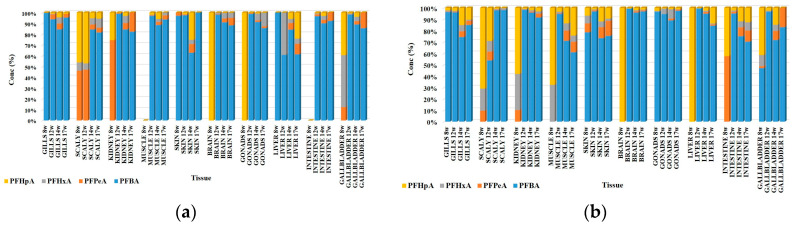
Distribution of PFOA metabolites in fish organs in Experiment 1 (**a**) and Experiment 2 (**b**).

**Figure 6 foods-12-01423-f006:**
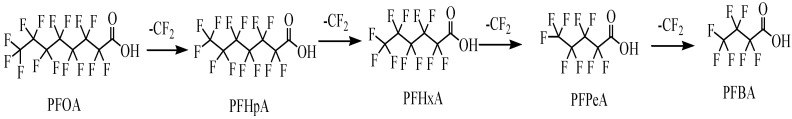
PFOA metabolization pathway in fish organs.

**Table 1 foods-12-01423-t001:** PFOA concentration values determined in the diluted water without fish in Experiment 1 and Experiment 2.

Time (h)	Concentration (µg/L) ± SD (*n* = 3)	
	10 µg/L	100 µg/L
0	10.0 ± 0.02	99.8 ± 1.85
1	9.99 ± 0.02	99.8 ± 1.85
24	9.98 ± 0.02	99.4 ± 1.85
48	9.92 ± 0.02	98.8 ± 1.83
72	9.89 ± 0.02	98.2 ± 1.82
96	9.76 ± 0.02	97.6 ± 1.81
Average	9.92 ± 0.02	98.9 ± 1.84
CV%	2.16	1.88

Note: The results are expressed as averages of 3 replicates ± SD.

**Table 2 foods-12-01423-t002:** BCF values determined in fish samples.

Test (Weeks, w)	Brain	Gonads	Intestine	Gallbladder	Kidney	Muscle	Liver	Gills	Skin	Scales
	Experiment 1 (10 µg/L)									
Exposure phase 8 w	24.5 ± 4.9	13.5 ± 2.2	8.86 ± 2.5	233 ± 20.4	39.4 ± 9.3	5.16 ± 0.8	49.2 ± 2.3	38.6 ± 1.5	28.8 ± 2.3	25.9 ± 2.7
12 w	28.4 ± 5.2	24.2 ± 1.3	25.4 ± 1.5	164 ± 18.4	44.6 ± 5.3	38.8 ± 1.3	14.8 ± 1.2	23.3 ± 1.3	35.7 ± 1.6	35.4 ± 3.9
14 w	2.31 ± 0.7	2.45 ± 0.5	7.66 ± 0.9	77.8 ± 10.2	34.5 ± 2.3	34.2 ± 1.4	3.32 ± 0.3	13.7 ± 0.3	32.6 ± 2.5	34.2 ± 1.8
Average of 8, 12, 14 w	18.4	13.4	14.0	158	39.5	26.0	22.4	25.2	32.4	31.8
Depuration phase 17 w	-	-	7.93 ± 1.3	64.7 ± 9.4	45.7 ± 1.4	0.12 ± 0.01	6.55 ± 0.6	15.5 ± 1.0	-	-
	**Experiment 2 (100 µg/L)**									
Exposure phase 8 w	14.4 ± 0.7	20.0 ± 1.3	6.78 ± 0.1	153 ± 10.34	111 ± 21.2	8.04 ± 0.7	22.3 ± 0.9	34.6 ± 2.5	20.7 ± 1.7	14.7 ± 3.2
12 w	10.1 ± 0.9	17.4 ± 2.4	10.3 ± 1.3	79.4 ± 12.4	20.2 ± 1.3	14.2 ± 0.4	6.75 ± 1.4	22.3 ± 1.8	15.9 ± 1.6	11.4 ± 2.1
14 w	0.68 ± 0.05	0.75 ± 0.02	0.83 ± 0.02	8.94 ± 0.8	1.57 ± 0.06	0.42 ± 0.006	0.98 ± 0.03	1.63 ± 0.7	0.87 ± 0.02	0.76 ± 0.04
Average of 8, 12, 14 w	8.4	12.7	5.97	80.3	44.3	7.55	10.0	19.5	9.48	8.97
Depuration phase 17 w	-	-	1.18 ± 0.05	10.3 ± 0.2	1.78 ± 0.6	0.14 ± 0.002	0.97 ± 0.01	1.56 ± 0.5	-	-

Note: The results are expressed as averages of four specimens.

## Data Availability

Data is contained within the article.

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
