# Peer review of "Tissue Bioconcentration Pattern and Biotransformation of Per-Fluorooctanoic Acid (PFOA) in Cyprinus carpio (European Carp)—An Extensive In Vivo Study"

_foods, 2023, doi:10.3390/foods12071423_

Round 1
Reviewer 1 Report
The authors investigated the bioconcentration and biotransformation of perfluorooctanoic acid (PFOA) in common carp (Cyprinus carpio). The results of this study will provide a better understanding of the ecotoxicological risks of PFOA. However, the results do not seem to be directly related to food safety. It is better to estimate the potential health risks to humans through consumption of common carp. Additionally, the authors should clearly elucidate the innovation of this study compared to previous studies in the section of introduction. Moreover, it should be noted the basis for setting the exposure concentration. Nevertheless, I think the paper is well written and can be accepted after revision.
Author Response
We would like to thank Reviewer 1 for his detailed reading of our manuscript and constructive comments, which we have used extensively in improving this article! The modified text is highlighted by ‘blue color/ track changes”.
Sincerely,
Corresponding author
Stefania Gheorghe (stefania.gheorghe@incdecoind.ro)
National Research and Development Institute for Industrial Ecology ECOIND, 57-73 Drumul Podu Dambovitei Street, 060652, Bucharest, Romania

Reviewer 2 Report
The paper entitled “Tissue bioconcentration pattern and biotransformation of Per-fluorooctanoic Acid (PFOA) in Cyprinus carpio (European Carp) - An extensive in vivo study” with manuscript ID 2227952 by Petre et al., performed an in-vivo analysis of the bioconcentration of PFOA in various tissues of European carp, as well as its biotransformation.
This study has shown that PFOA can be bioconcentrated in all of the tissues examined and that this bioconcentration is significant and depended on the duration and the magnitude of the exposure. However, before this study can be considered for publication there are few points that should be addressed.
First, the authors have not provided any information regarding the rearing conditions of the fish prior to experimentation. Moreover, during experimentation the water temperature showed large variation, and potential effects of this fact should be discussed by the authors.
Second, no statistical analysis on the results has been performed when comparing the bioconcentration of the different tissues under study. Moreover, no data on the variation of the results is presented either in the Figures or the Tables. Finally, the correlations tests have been performed using Pearson’s correlation, while the authors have not provided information regarding the normality of the data or the possible violation of the test’s assumptions.
Specific comments
Introduction
L45-77: This paragraph is very large and includes a lot of different subjects. In my opinion, the MS would benefit from a distinction of the different subjects in different paragraphs.
L50-51: Please elaborate on how it functions as an endocrine disruptor.
L83-88: Please reduce the size by including only the most relevant results. This information seems quite extensive for an introduction.
L93: I would recommend that the authors change the paragraph in this place.
L105-109: Please rewrite the objectives of the study. More details are needed, as for example that the study was conducted on common carp and in fish in general.
Materials and Methods
L178-180: Please add information regarding the rearing of the fish. Tank size, fish density, water temperature and other essential quality parameters, type of water renewal (RAS or open-flow) are essential information that should be presented.
L181-183: Please provide more information on the feeding. Proportion of substances in the feed and most importantly how was feeding performed. By hand? One, twice or more times a day? Was it performed on every day of the week?
L194: How were fish sacrificed? Using anesthesia? If so, what type and concentration?
L225: This is a very wide temperature range and should be discussed for possible effects on the results.
Results & Discussion
L249: This is the first mention of Experiments 1 and 2 as such, and I would therefore recommend, for readers facilitation, that the authors distinguish these experiments more clearly in the Materials and Methods Section.
Fig1d: Please rephrase "Skaly" to "Scales".
Figure: Not sure what "Martor" refers to. Please provide more information in the figure caption.
L368: The number of this citation is missing.
Table 3. The variation statistics are missing. Please include the SD for each measurement. The same comment applies for the figures as well.
L394: How can the authors rank the concentrations when no statistical comparisons have been performed?
L464: Pearson.
L467: Pearson correlation. according to its assumptions, should be used only when the data are from a population the follows a normal distribution. Have the authors checked this assumption?
L468: Why is it “≤” and not “=”? The same comment applies for all correlation results.
Author Response
We would like to thank Reviewer 2 for his detailed reading of our manuscript and constructive comments, which we have used extensively in improving this article! The modified text is highlighted by ‘blue colour / track changes”.
Sincerely,
Corresponding author
Stefania Gheorghe (stefania.gheorghe@incdecoind.ro)
National Research and Development Institute for Industrial Ecology ECOIND, 57-73 Drumul Podu Dambovitei Street, 060652, Bucharest, Romania

Reviewer 3 Report
This study aims to present a vivo study on the tissue bioconcentration pattern and biotransformation of PFOA in Cyprinus carpio. Generally, the manuscript is well-written and considers all aspects. A suggestion is that the authors should express all average values with standard deviation, while the bar chart should be provided with an error bar. Besides, a sub-section should be added to declare the statistical methods applied in the data treatment.
Author Response
We would like to thank Reviewer 3 for his detailed reading of our manuscript and constructive comments, which we have used extensively in improving this article! The modified text is highlighted by ‘blue colour/ track changes”.
Sincerely,
Corresponding author
Stefania Gheorghe (stefania.gheorghe@incdecoind.ro)
National Research and Development Institute for Industrial Ecology ECOIND, 57-73 Drumul Podu Dambovitei Street, 060652, Bucharest, Romania

Round 2
Reviewer 2 Report
The authors have addressed the vast majority of the comments and have enhanced the outcome of the paper.
There is still, however, an issue that the authors have not responded accordingly and are highly encouraged to respond to. Pearson correlation should only be used were the examined data are derived from populations that follow a normal distribution. The authors, as far as I can tell from their response to my previous comment, have not perform a normality test before using the Pearson correlation. It is crucial that the authors perform normality test before using the Pearson correlation. If normality of the data is observed then they can use Pearson's test. If not, they should use the non-parametric Spearman correlation which doesn't assume normality of the data.
Author Response
We would like to thank Reviewer 2 for his constructive comments, which we have used extensively in improving this article!
We replaced the Pearson correlation with the Spearman parameter, redid all the calculations and replaced the corresponding values both in the manuscript and in the supplementary material. The obtained results generated values not very different from those obtained for Pearson correlation.
The changes were performed in track changes.
Thank you,
Gheorghe Stefania